# AgriEnt: A Knowledge-Based Web Platform for Managing Insect Pests of Field Crops

**Katty Lagos-Ortiz [1], María del Pilar Salas-Zárate [2], Mario Andrés Paredes-Valverde [2]** **,**
**José Antonio García-Díaz [3] and Rafael Valencia-García [3,*]**

[1] Agrarian University of Ecuador, Guayaquil 090104, Ecuador; klagos@uagraria.edu.ec
[2] Division of Research and Postgraduate Studies, Tecnológico Nacional de México/I. T. Orizaba,
   Orizaba 94320, Mexico; msalasz@ito-depi.edu.mx (M.d.P.S.-Z.); mparedesv@ito-depi.edu.mx (M.A.P.-V.)
[3] Departamento de Informática y Sistemas, Facultad de Informática, Universidad de Murcia,
   Campus de Espinardo, 30100 Murcia, Spain; joseantonio.garcia8@um.es
[*] Correspondence: valencia@um.es; Tel.: +34-868-888522

**Abstract:** In the agricultural context, there is a great diversity of insects and diseases that affect crops. Moreover, the amount of data available on data sources such as the Web regarding these topics increase every day. This fact can represent a problem when farmers want to make decisions based on this large and dynamic amount of information. This work presents AgriEnt, a knowledge-based Web platform focused on supporting farmers in the decision-making process concerning crop insect pest diagnosis and management. AgriEnt relies on a layered functional architecture comprising four layers: the data layer, the semantic layer, the web services layer, and the presentation layer. This platform takes advantage of ontologies to formally and explicitly describe agricultural entomology experts' knowledge and to perform insect pest diagnosis. Finally, to validate the AgriEnt platform, we describe a case study on diagnosing the insect pest affecting a crop. The results show that AgriEnt, through the use of the ontology, has proven to produce similar answers as the professional advice given by the entomology experts involved in the evaluation process. Therefore, this platform can guide farmers to make better decisions concerning crop insect pest diagnosis and management.

**Keywords:** ontology; agriculture; insect pests

## 1. Introduction

Agriculture plays a critical role in the economy of many countries, since it contributes, either in a small or big way, to the production of essential food crops, employment generation as well as national income. The agricultural industry faces challenges such as water shortages, soil fertility, pests and diseases affecting crops, and increasingly rigorous standards on quality and safety food, to mention but a few. Regarding crop pests, these represent one of the major constraints to increase food production [1] since they severely damage crop plants and reduce the quality of food grains and products, which causes considerable economic losses [2]. From this perspective, if invasive pests and pathogens continue spreading, their potential costs to global agriculture could be more than 540 billion dollars per year [3]. An insect pest can be referred to as any insect species whose activities, enhanced by population density, causes economic losses to cultivated crops [4]. Nowadays, there are different methods for insect pests controlling such as good farming practices, breeding and growing resistant varieties, biological control, among others. However, the use of chemical pesticides is the main mode of pest control among farmers [5], which severely increases environmental pollution as well as contributing to the reduction of the population of natural enemies of herbivores [6].

In the agricultural context, there is a great diversity of insects and diseases that affect the crops. Moreover, the amount of data available on data sources such as the Web regarding these topics increase every day. This fact can represent a problem when farmers want to make decisions based on this large and dynamic amount of information. From this perspective, to diagnose the insect pest that is affecting a crop is a serious challenge to farmers since this task demands knowledge and the experience of experts on insect pests. Therefore, there is a need for systems that integrate an expert's knowledge to support farmers to perform diagnosis, control, and management of insect pests aiming to improve the quality of food grain as well as to reduce economic losses.

The Semantic Web provides to the Web information with a well-defined meaning which allows humans and computers to understand it. One of the pillars of the Semantic Web are the ontologies, which can be referred as to a formal and explicit specification of a shared conceptualization [7]. Nowadays, decision support systems based on ontologies have been widely accepted as solutions in different domains such as recommender systems [8], software engineering [9], and health [10–12]. This opens the door for opportunities to apply this technology to the agriculture domain, specifically for crop insect pest's diagnosis and management.

This work presents AgriEnt, a knowledge-based Web platform focused on supporting farmers in the decision-making process concerning crop insect pest diagnosis and management. AgriEnt takes advantage of ontologies to formal and explicit describe the agricultural entomology experts' knowledge and to perform insect pest diagnosis through a rule-based inference engine that recommends the appropriate treatments to deal with the diagnosed insect pest, considering the symptoms provided by the user. By using an ontology, it is possible for non-expert people to take advantage of the knowledge of experts on this domain to decide which insect pest is damaging their crops. Furthermore, the Web application provided by this platform makes it possible for farmers to get decision support regardless of where they are since it can be accessed from any device with internet access and a Web browser installed. This paper describes the AgriEnt platform, emphasising the modeling of the ontology for crop insect pest management as well as the implementation of rules based on SWRL (Semantic Web Rule Language) [13] for diagnosing a crop insect pest based on the symptoms provided by the user. It should be mentioned that this work represents an extension of previous work by the authors [14]. However, the present work describes a great amount of new content with respect to such work; specifically, it presents the following new contributions: (1) the ontology was extended to include other crops such as cocoa and banana, as well as the insects pests that affect them; (2) a reliability evaluation was performed in order to know the acceptance of the application by the final users; and (3) the performance of the system regarding insect pest diagnosis was improved compared with the previous work.

This work is structured as follows: Section 2 describes the relevant literature on ontology-based decision support systems in different domains. Then, Section 3 describes the functional architecture of AgriEnt, highlighting those modules and resources used for crop insect pest diagnosis purposes. Section 4 presents a case study regarding insect pests' diagnosis, specifically, it describes a quantitative evaluation focused on measuring the accuracy of AgriEnt regarding insect pest diagnosis and a user-centered evaluation focused on measuring perceived usefulness and usage intentions of farmers involved in this work regarding the AgriEnt platform. Finally, Section 5 presents conclusions and future research directions.

## 2. Related Work

Ontologies have been widely adopted for developing decision support systems in different domains. For instance, in the medical context, there are decision support systems that assist professionals in their daily activities such as disease diagnosis, care given, intensive-care medicine, antibiotic management, among others, thereby improving the quality of life of patients. OntoDiabetic [11] is an ontology-based decision support system that assesses health risk factors of patients with diabetes, cardiovascular disease, and hypertension, and suggests appropriate treatments. To this end, OWL2 (Web Ontology

Language) rules are applied to a semantic patient profile that collects information concerning vital signs, nutritional history, and laboratory test values. In [12], the authors presented a recommender system focused on antidiabetic medication. This system considers the safety and positivity of HbA1c to determine the ranking of antidiabetic medications. To this end, it uses medication profiles, a drug knowledge ontology and the TOPSIS (Technique for Order of Preference by Similarity to Ideal Solution) technique for calculating the ideal solution. In [15], the authors presented an automated reasoning methodology to assign alleles and phenotypes to patients and to match patients to appropriate pharmacogenomic guidelines. This methodology works over ontologies that provide a formal representation of pharmacogenomic knowledge. In [16], a clinical decision support system for the antibiotic stewardship program is presented. This system implements production rules, ontologies, and workflow modelling techniques to provide a multi-user perspective and reactive and proactive behaviors. Finally, SHKB (Semantic Healthcare Knowledge Base) [17] is a semantic-based approach to represent healthcare domain knowledge and patient data aiming to provide support to clinical decision making. This approach implements an ontology that follows the HL7 (Health Level Seven) clinical document architecture [18].

Knowledge-based decision support systems also have been adopted in domains other than health. For instance, in [19], the authors presented an ontology-based decision support tool that helps users to select a domestic solar hot water system that meets the user's needs such as installation costs, components and their interrelationships, number of occupants, house location, and daily hot water requirements. In [20], an ontology-enabled decision support system for manufacturing process selection is described. This system helps manufacturing engineers to determine appropriate processes to design products with a competitive matching between features, material characteristics, and process capabilities. In [21], the authors proposed a decision support framework for the prefabricated component supply chain. This framework uses ontologies for providing unified support for the simulation process as well as for integrating heterogeneous data sources. In [22], the authors presented a monitoring and decision support system for engineering vehicles. This system implements an ontology that models monitored rollover stability data. Also, SWRL-based rules were developed to assess rollover risk and obtain suggested measures. In [23], the authors proposed an ontology-based environmental decision support system that integrates data from different Web data sources and assesses these fused data for a given time and location. In this way, users can interpret information and make a decision. In [24], the authors described a framework for intelligent environmental decision support systems. This framework integrates ontologies, data mining, and Bayesian networks to generate a knowledge base that is used for decision-making purposes. In addition, some research has been performed to ease the use of ontologies for non-expert users. In [25], the authors presented OWL-Path, a natural language-query editor guided by ontologies to ease non-experts users with the creation of SPARQL (SPARQL Protocol and RDF Query Language) queries. OWL-Path was evaluated under different domains such as e-finance and e-tourism. Along the same lines, ONLI (Ontology-based Natural Language Interface) [26] allows non-expert users to query DBPedia with natural language, inferring the answer type expected by the user through an established question's classification. This system makes use of ontologies in order to represent both the syntactic question's structure and the question's context.

Historically, agriculture has been benefitted from technology adoption. In the last few years, there have been numerous research efforts to provide ontology-based technological solutions for insect pest control and management. For instance, in [27], an ontology-based prototype for planning intercropping is presented. This system uses an optimization model that takes into account the farmer's constraint factors to illustrate maximum income as well as to minimize the cost of plant cultivation. In [28], the authors presented an ontology-based expert system for managing pests and diseases that affect grape crops from India. This system generates a knowledge base about grape pests and diseases from Web data sources and implements fuzzy logic rules that consider weather conditions for forecasting probable pests and diseases. SePeRe (Semantic Pest Recognition) [29] is a system for the early detection

of plant diseases and pests which relies on a relational database that contains information about diseases and pests of Mediterranean crops such as olives, grapes, almonds, among others. This system receives as input an image of the crop and provides information about the disease affecting the plant.

Most of the works previously analyzed present ontology-based decision support systems focused on the medical domain or, to a lesser extent, other domains such as manufacturing, environment, supply chain, and engineering vehicles. Specifically, on the identified systems focused on the agriculture domain, we have found that, none of them consider the diagnosis of crop insect pests. Table 1 presents a comparison of the most relevant works previously discussed.

**Table 1.** Comparative analysis of ontology-based decision-support systems.

| Work | Objective | Domain | Techniques Used for Rule-Based Engine |
|------|-----------|--------|---------------------------------------|
| [11] | Suggest diabetes treatments | Diabetes | OWL2 |
| [12] | Ranking of antidiabetic medications | Diabetes | Fuzzy rules |
| [16] | Antibiotic management | Clinical | Drools (Business Rules Management System) |
| [17] | Unify representation of healthcare domain knowledge and patient data | Healthcare | Jena |
| [19] | Optimizing domestic solar hot water system selection | Energy saving | Jena |
| [20] | Manufacturing Process Selection | Manufacturing | SWRL |
| [21] | Supporting the prefabricated component supply chain | Supply chain | Jena |
| [22] | Rollover Monitoring and Decision Support System for Engineering Vehicles | Rollover | SWRL |
| [23] | Provide environmental information for personalized decision support | Environmental domain | LSR (Logico-Semantic Relations), OWL-DL |
| [24] | Generating valuable environmental knowledge | Environmental domain | Fuzzy rules |
| [27] | Minimizing the cost of cultivating the plants. | Intercropping | Jena |
| [28] | Reduce loss in grape yield. | Grapes | Fuzzy rules (jfuzzylogic) |

As can be seen from Table 1, ontologies have been implemented in decision support systems in different domains. However, relatively little work has been done in the development of decision support systems for insect pest diagnosis. For instance, works such as that presented in [28] are focused on only one crop, in this case, the grape. On the other hand, despite all works analyzed make use of ontologies as the main mechanism to represent expert's knowledge, they differ in the techniques used for developing the rule-based inference engine. Research efforts such as the presented in [17,19,21,27] implemented rule-based inference engines that rely on Jena, a Java-based application framework for developing semantic web applications. Jena provides development tools and predefined reasoners including an RDFS (Resource Description Framework Schema) reasoner, and OWL-Lite reasoner, and a generic rule reasoner that supports user-defined rules written in the Jena rules language. Despite all these advantages, Jena provides a lower degree of expressiveness of the rule language in comparison with rule language such as SWRL [30]. Specifically, Jena rule language does not provide support for conjunction, disjunction and negation operations as well as universal and existential logical quantifiers. This fact limits the expressiveness that can be required to define complex rules that help to provide support in decision tasks from different domains. Meanwhile, for rule definition and inference purposes, in [16], the authors used Drools, an open-source software that provides a core business rules engine (BRE), a web authoring and rules management application (Drools Workbench). Despite Drools

is a good option to develop rule-based decision support systems, it does not allow the direct use of ontologies. Furthermore, Drools adopt an object-oriented approach, which is less expressive than OWL, which has a better representation and inference [31]. Also, in Drools, rules are written in DRL (Drools Rule Language) language, an open-source but non-standard language. In this sense, SWRL rules are simpler and easier to define than DRL rules [31]. On the other hand, works such as [12,24,28] relied on fussy technologies for developing the rules. Like the Drool tool, fuzzy technologies used in these works (such as jfuzzylogic) do not allow the direct use of ontologies. From this perspective, this work adopts SWRL, the most commonly used language used to express rules [32], for defining the rules used for insect pest diagnosis purposes. SWRL-based rules work directly with the concepts and relationships defined in the OWL-based ontology model. Therefore, adaptation would be represented explicitly in the ontology. Finally, despite there being works that have successfully used SWRL for defining rules that support decision tasks, they are not focused on insect pest diagnosis. In the following section, the main components of AgriEnt and their interrelationships are described in detail.

## 3. AgriEnt: Architecture

AgriEnt is a knowledge-based decision support system that helps farmers to control and manage crop insect pests. It is widely recognized that for managing and analyzing information for the decision-making process, a decision support system must consist of at least three main elements: (1) a user interface, (2) a database, and (3) a model processing module [33]. From this perspective, AgriEnt relies on a layered functional architecture comprising four layers (see Figure 1): (1) the data layer, (2) the semantic layer, (3) the web services layer, and (4) the presentation layer. Each layer, in turn, consists of several components with well-defined functions and responsibilities thus enabling the platform to be scalable and easy to maintain.

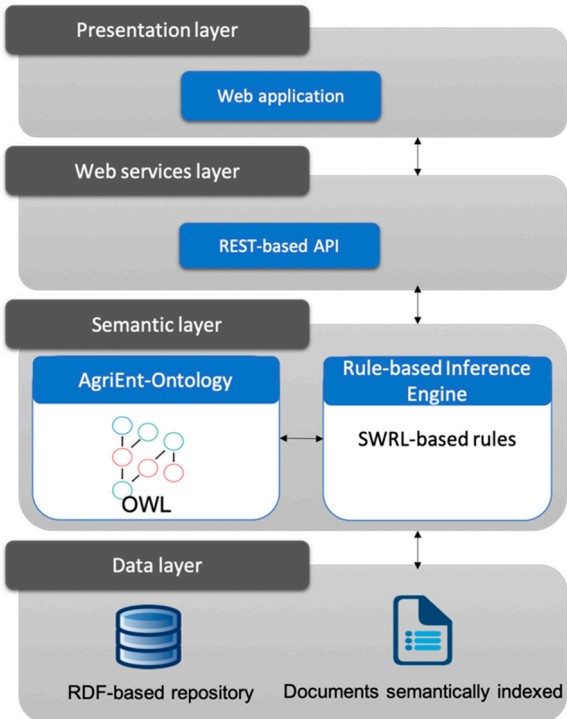

**Figure 1.** General architecture of AgriEnt.

Figure 1 depicts the general architecture of AgriEnt whose general operation is as follows: farmers provide through a Web application a set of symptoms they gathered by observing the crops they are in charge of. Based on the symptoms provided, the rule-based inference engine provides a diagnosis about the insect pest affecting the crop and provides recommendations with the appropriate treatments

to deal with it. Furthermore, AgriEnt extends this information with a set of documents about the control and management of that insect pest aiming to allow users to increase their knowledge about this phenomenon. The communication between the Web application and the semantic layer is performed through the web services layer, which provides a REST-based (Representational State Transfer) API (Application Programming Interface) that is in charge of processing all incoming requests. The layers and components of AgriEnt are thoroughly described below.

### 3.1. Data Layer

A semantic Web-based knowledge base consists of a T-Box layer, also referred to as ontology, and an A-Box layer, which consists of instances of concepts described by the T-Box layer. The A-Box layer of AgriEnt consists of a resource description framework (RDF)-based repository of instances of crops, diseases, symptoms, insects, insect pests, and treatment recommendations. This layer was developed from crop insect pests' records generated by agricultural entomology experts from Ecuador. These records contain information about the insect pests, treatments followed, dates and specific places on which they occurred, among other facts.

On the other hand, AgriEnt not only aims to support insect pests' control and management but also increases farmers' knowledge about insect pests' control and management as well as alternatives to chemical pesticides. To this end, this platform provides users with documents related to these topics. The document database used by AgriEnt contains research works from the Agrarian University of Ecuador as well as free-available academic publications. All these documents were semantically indexed by using the semantic annotation approach proposed in [34]. This approach consists in two main phases. Firstly, all documents are semantically annotated in accordance with the AgriEnt-Ontology, and secondly, a weight is assigned to each annotation to determine how relevant is a concept described by the ontology for the document meaning. The weighting process follows the adaptation of the vector-space model for ontology-based information retrieval presented in [35]. The weighting mechanism implemented by this platform considers the concepts that explicitly appear in the documents as well as all those concepts that have a taxonomic relation with them. To achieve this goal, the Dijkstra algorithm [36] is used to find the shortest path between two concepts of an ontology.

### 3.2. Semantic Layer

The semantic layer relies on the AgriEnt-Ontology (Agricultural Entomology) and a rule-based inference engine. On the one hand, the role of the AgriEnt-ontology is to capture expert's knowledge about a crop insect pest's management as well as to provide an agreed-upon understanding of such a domain. On the other hand, the rule-based inference engine examines the crop's symptoms provided by the user against a set of rules to provide a diagnosis about the insect pest affecting the crop. Both ontology and rule-based engine components are widely discussed below.

#### 3.2.1. AgriEnt-Ontology

One of the most important phases of the development of a knowledge-based system is the development of a domain ontology [37]. From this perspective, this platform uses the AgriEnt-Ontology to represent knowledge about crops, diseases, symptoms, insects, insect pests, and treatment recommendations. This knowledge is represented in a formal way consisting of concepts, relationships, individuals, and axioms. In this way, the ontology can be shared and reused whenever necessary. The ontology was modelled by using Protégé [38], an ontology development platform that allows editing ontologies in the Web Ontology Language (OWL), accessing description logic reasoners, as well as acquiring instances for semantic markup. The ontology development process followed the Methontology [39] knowledge engineering methodology. An excerpt from the AgriEnt-Ontology is illustrated in Figure 2.

The knowledge described by this ontology was collected from crop insect pests' records generated by agricultural entomology experts as well as academic publications. Furthermore, the design of

this ontology considered already available ontologies such as (1) the Plant Disease Ontology [40], (2) the Ontology for Plan Protection [41], and (3) a plant disease extension of the IDO (Infectious Disease Ontology) ontology [42]. The Plant Disease Ontology describes information about the anatomy, morphology, genomics and proteomics of plants. Meanwhile, the Ontology for Plan Protection describes activities related to cereal plan protection. The IDO ontology covers infectious plant diseases and pathogens. Finally, it is worth noting that the AgriEnt-Ontology was validated by researchers from the Agrarian University of Ecuador with experience on insect pest's management.

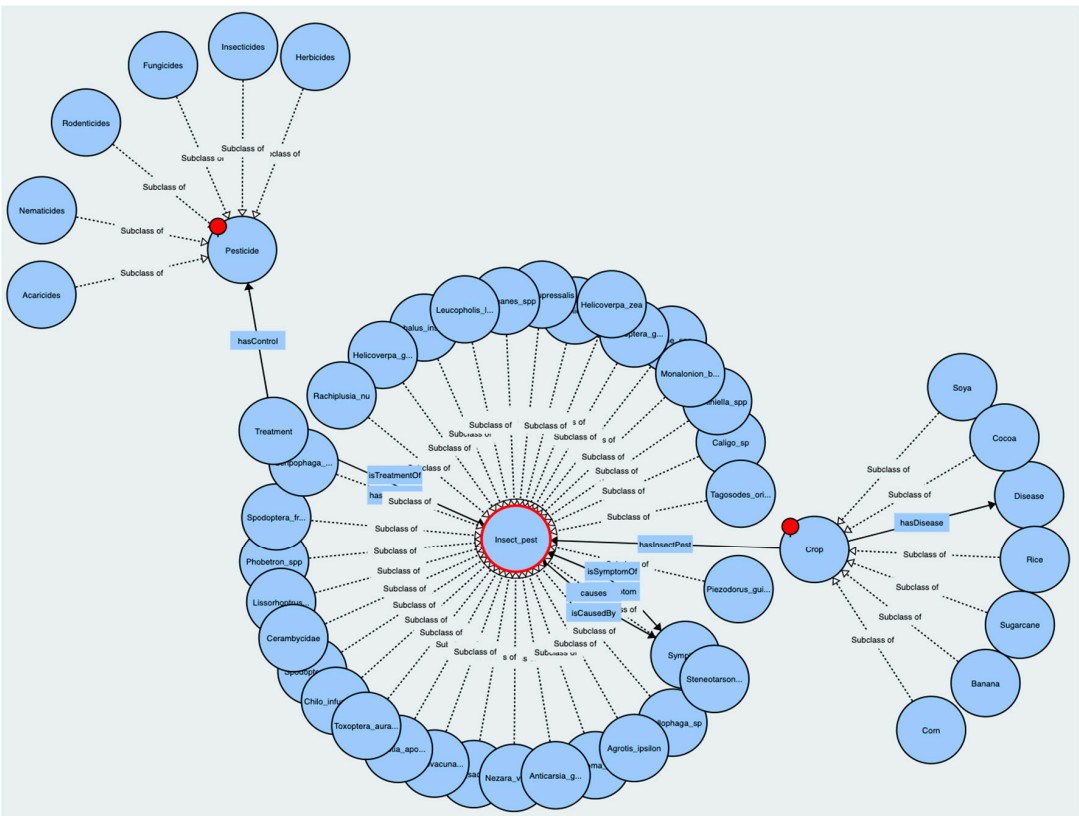

**Figure 2.** An excerpt from the AgriEnt-Ontology.

Table 2 describes the main classes on the AgriEnt-Ontology and provides some examples of instances contained by the knowledge base.

**Table 2.** Classes defined by the AgriEnt-Ontology.

| Concept | Description | Instances Examples |
|---|---|---|
| Crop | It refers to a taxonomy of cultivated plants that grown as food, especially a grain, fruit, or vegetable. | Sugarcane, cocoa, corn, rice, banana, and soy. |
| Insect pest | It refers to a taxonomy of insects that reduce yields and spread virus diseases by feeding on the plants. | *Chilo infuscatellus*, *Scripophaga excerptalis*, *Pyrilla perpusilla* |
| Symptom | It refers to a taxonomy of symptoms i.e., phenomes accompanying something and is regarded as evidence of its existence [24]. | Dead heart in 1-3 months old crop; Red tunnels in the midribs of leaves; Affected tissues reddened |
| Disease | It refers to a taxonomy of diseases that affect crops. | Red Rot (*Colletotrichum falcatum*), Smut (*Ustilago scitaminea*), Pineapple Disease (*Ceratocystis paradoxa*) |
| Treatment | It consists of a taxonomy of treatments for controlling insect pests. | Release Ichneumonid parasitoid; Provide adequate irrigation; Crop rotation in endemic areas |

The main classes defined by AgriEnt-Ontology are Crop, Insect pest, Symptom, Disease, and Treatment. All these classes are disjoint classes to prevent an individual from being defined as an instance of more than one class. Currently, the ontology contains information about common crops in Ecuador such as sugarcane, cocoa, corn, rice, banana, and soy; as well as common insect pests that affect them. Also, this ontology describes diseases that can be caused by the such insect pests. Finally, this ontology collects treatments to deal with insect pests. These treatments were validated by a group of domain experts. AgriEnt-Ontology defines a set of object properties that allows relations between the classes to be established. Through these properties is possible to perform the crop insect pest diagnosis as well as to provide suitable treatments. Table 3 describes the main object properties including their domain, range, and use. The hasSymptom property represents which symptoms are associated with specific insect pests. An insect pest can be diagnosed based on different symptoms and a symptom can be associated with multiple insect pests. In addition to the properties described in Table 3, AgriEnt-Ontology defines the inverse object properties of hasSymptom-isSymptomOf, isCausedBy-causes, and hasTreatment-isTreatmentOf.

Finally, it must be mentioned that the ontology described above is scalable, therefore new crops, symptoms, diseases, and insect pests can be added without affecting the overall functionality of the platform.

**Table 3.** Properties associated with the classes defined by AgriEnt-Ontology.

| Property Name | Domain | Range | Use |
| --- | --- | --- | --- |
| hasSymptom | Insect pest | Symptom | It associates an insect pest with their symptoms. |
| isCausedBy | Symptom | Insect pest | It associates a symptom with the insect that cause it. |
| hasTreatment | Insect pest | Treatment | It associates an insect pest with suitable treatments. |

### 3.2.2. Rule-Based Inference Engine

The farmer's tasks include harvesting and inspecting crops, irrigating farm soil, spraying fertilizer, among others. However, one of the most critical tasks is to diagnose the disease that is affecting their crops. This task consists in identifying the cause of the disease by analyzing the signs and symptoms of their crops. AgriEnt is designed to assist this task through an inference engine that examines the crop's symptoms provided by the farmer against the set of rules, which are used to represent experts knowledge in intelligent systems [30]. These rules were defined by using SWRL, one of the most used languages to express rules on semantic-based systems [32]. SWRL allows rules to be defined in terms of OWL concepts to provide deductive reasoning capabilities. Besides, these rules increase the level of expressivity that is not provided by the OWL language i.e., they allow defining relations that cannot be defined through OWL DL (Description Logic).

The rules definition process involved a group of agricultural entomology experts who were interviewed about diseases, insect pests, symptoms, and recommendations of common insect pests in Ecuador. Then, they were asked to establish a set of conditions focused on diagnosing insect pests based on a set of symptoms. Once these conditions were defined, the authors expressed these conditions in the form of if-then statements consisting of an antecedent, also known as the body of the rule, and a consequent, which is also referred to as the head of the rule. Specifically, all rules follow the format depicted in Equation (1):

$$R_1, R_2, \ldots, R_n \rightarrow D \tag{1}$$

where $R_1, R_2, \ldots, R_n$ is the antecedent part, which contains atomic formulas that represent conditions, while D is the consequent part, which contains conclusions i.e., is the conclusion when all conditions

from the antecedent part are fulfilled. For instance, Equation (2) presents the definition of the *Scripophaga excerptalis* sugarcane pest:

$$
\begin{aligned}
&pest(?x) \wedge hasSymptom(?x, \text{"parallel rows of shot holes in the emerging}\\
&\text{leaves"}) \wedge hasSymptom(?x, \text{"red tunnels in the midribs of leaves"}) \wedge hasSymptom(?x,\\
&\text{"dead heart in grown up canes"}) \wedge hasSymptom(?x, \text{"dead heart reddish brown in}\\
&\text{color"}) \wedge hasSymptom(?x, \text{"bunchy top due to growth of side shoots"}) \rightarrow isPest(?x,\\
&\text{"Scripophaga excerptalis"})
\end{aligned}
\tag{2}
$$

The rule presented in Equation (2) specifies that the Scripophaga excerptalis sugar cane pest will be inferred by the inference engine when all symptoms specified by the hasSymptom conditions match with the symptoms of such insect pest, in this case, parallel rows of shot holes in the emerging leaves, red tunnels in the midribs of leaves, dead heart in grown up canes, dead heart reddish brown in color, and bunchy top due to growth of side shoots.

Once insect pest is diagnosed, AgriEnt provides users with information that helps them to better identify the life stage of the insect (egg, larva, pupa, and adult). Sometimes, such information can influence the treatment to follow. Moreover, AgriEnt also provides a set of recommendations for controlling and managing the insect pests diagnosed. For instance, for the *Scripophaga excerptalis* pest, the platform provides the recommendations presented in Table 4. Aiming to provide such suggestions, the platform uses the object property hasTreatment described in Table 3.

**Table 4.** Treatment suggestions for *Scripophaga excerptalis*.

| | Treatment Suggestions |
|---|---|
| 1 | Collect and destroy the egg masses. |
| 2 | Release Isotima Javensists at 100 pairs/ha |
| 3 | Spray insecticides such as Carborufan 3% G 33.3 kg/ha, Phorate 10% G 30 kg/ha, or Chlorantraniprole 18.5% SC 375 mL/ha |

### 3.3. Presentation Layer

In many cases, farmers make their decisions solely on advice received by experts. However, in rural areas, there is a lack of agricultural experts that provide the support required. To address this issue, the AgriEnt platform provides a Web application that farmers can use to make a decision regarding insect pest control and management by providing the crop's symptoms as input and obtain a diagnosis of the insect pest affecting their crops. One of the main advantages of Web applications is that they can be used in every device with internet access and a Web browser installed. Furthermore, the Web application follows a responsive design, i.e., all Web pages render and work well across desktops and mobile devices such as tablets and smartphones.

Figure 3 depicts the main user interface of the AgriEnt Web application which contains a menu with the available options such as diagnosis, prevention, and login. The diagnosis Web page shows the six different crops for which AgriEnt provides support for insect pest management. Once users select the crop, the application asks them for all symptoms they perceived in their crop. Then, based on all symptoms specified by the farmer, the Web application shows the diagnosed insect pest, the reason why it was selected, and a set of recommendations for controlling and managing it. On the other hand, the Web application is also valid for prevention purposes i.e. it also provides users with a set of expert recommendations for preventing insect pests.

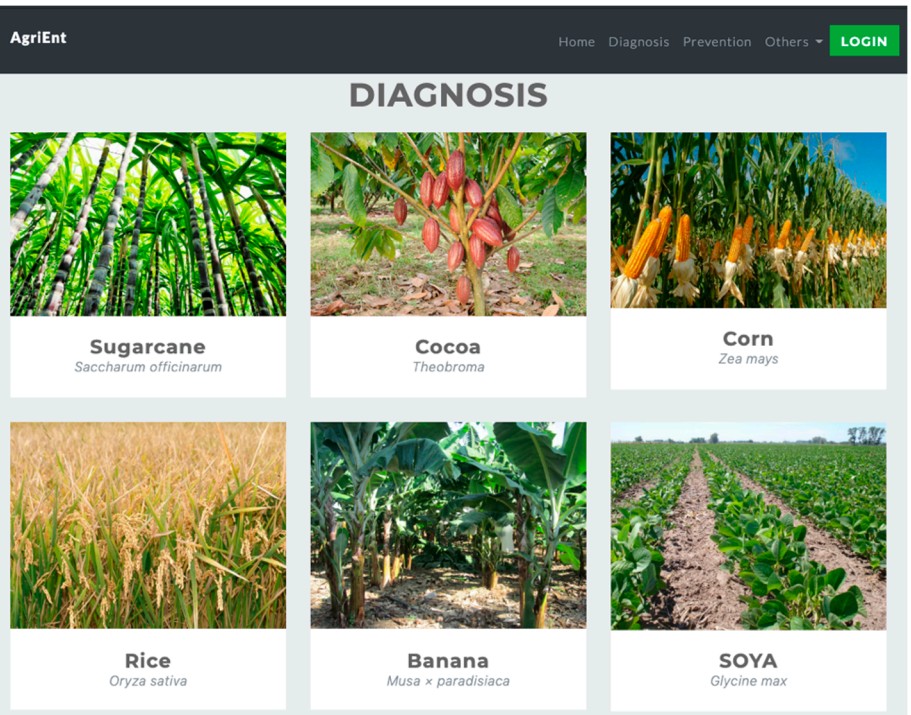

**Figure 3.** Main user interface of the AgriEnt Web platform.

## 4. Case Study: Diagnosing the Insect Pest Affecting a Crop

### 4.1. Methodology

This section describes the case study performed to validate the contribution of the AgriEnt platform to the control and management of crop insect pests. This case study focused on diagnosing the insect pest affecting a crop based on a set of symptoms gathered by the farmer. To this end, this methodology was followed:

1. People involved in selection. Twenty people from the Costa Region of Ecuador with experience in the managing of crops were asked to collect all symptoms they visually perceived when an insect pest is affecting their crops. The distribution of age of people involved ranged from 32 to 45 years old. These people needed to have been in charge of sugarcane, cocoa, corn, rice, banana, or soya field crops during two or more years. This requirement was necessary, so people were able to detect most symptoms. Since this case study involves humans, it was necessary that farmers being aware they were being just collaborating to evaluate the AgriEnt platform and the evaluation results would not affect their work.

2. Providing symptoms through the Web application. The AgriEnt Web application was introduced to the people involved in this case study. Then, they were asked to provide symptoms through this application. For instance, a farmer in charge of rice crops provided the system the following symptoms: laceration of the tender leaves, yellow streaks on the leaves of young seedlings, nursery and main field damaged, and terminal rolling and drying of leaves from tip to base. This task was performed over a period of four months (February–May 2019) during which 149 sets of symptoms were collected. A description of the sets of symptoms collected is provided in Table 5.

3. Crop insect pest diagnosis. Each time users provided a set of symptoms through the Web application, the AgriEnt platform provided a diagnosis about the insect pest. For instance, considering the set of symptoms described in the previous point, the platform diagnosed the *Stenchaetothrips biformis* insect pest. As has been mentioned, to perform this diagnosis, the platform considers the knowledge modeled by the ontology as well as the rules defined. Similar

processes were performed for all set of symptoms provided by the farmers which refer to all crops considered by the ontology.

4. Measuring the accuracy of AgriEnt regarding insect pest diagnosis. Once all insect pest diagnostics were performed by the platform, these ones were compared with the diagnosis provided by the group of professionals involved in this case study. Then, the accuracy of AgriEnt regarding crop insect pest diagnosis was calculated by using Equation (3).

$$\text{Accuracy} = C/A * 100 \tag{3}$$

where C refers to the test cases where AgriEnt produced a correct insect pest diagnosis. Meanwhile, A refers to all test cases that were provided as input to AgriEnt for diagnosis purposes. The next section presents and discusses the results obtained by AgriEnt.

### 4.2. Results and Discussion

The rule-based inference engine allows AgriEnt to analyze crop symptoms collected by farmers aiming to contribute to the insect pest diagnosis and their management. The set of crop symptoms used in this case study were analyzed to gain insight into the main causes of incorrect diagnosis performed by the inference engine. The obtained results and the findings and recommendations of the evaluation are below described.

Table 5 presents the results obtained by AgriEnt concerning the diagnosis of crop insect pests. On average, AgriEnt obtained a 0.8221 score of accuracy in the reasoning of crop insect pest diagnosis, thus producing correct diagnosis in 123 test cases for all crops considered in this version. As can be noted, there is no big difference between the accuracy scores obtained for all crops considered in this case study. The accuracy scores range from 0.7857 to 0.875, specifically, the AgriEnt platform obtained the highest accuracy score (0.875) diagnosing insect pests that affect corn crops. On the other hand, this platform obtained the lowest accuracy score (0.7857) for insect pests affecting rice crops.

**Table 5.** Test results—diagnosis of crop insect pests.

| Test | Crop | Test Cases | Correct Test Cases | Accuracy |
|------|------|------------|--------------------|----------|
| 1 | Sugar | 24 | 20 | 0.8333 |
| 2 | Cocoa | 19 | 15 | 0.7894 |
| 3 | Corn | 32 | 28 | 0.875 |
| 4 | Rice | 28 | 22 | 0.7857 |
| 5 | Banana | 25 | 21 | 0.84 |
| 6 | Soya | 21 | 17 | 0.8095 |
| Total | | 149 | 123 | 0.8221 (Avg.) |

As can be noted from Table 5, a similar diagnosis was performed by both the group of professionals and the inference engine implemented in AgriEnt. However, this platform could not provide a correct diagnostic in 26 test cases thereafter due to the following facts:

- Most of the insect pest cases (11 test cases) were incorrectly diagnosed since farmers provided a small number of symptoms, which were not enough to determine the correct insect pest. Regarding this fact, most of these cases were related to the lack of experience of farmers in collecting sufficient crop symptoms.
- The remaining incorrect diagnosis cases were due to incorrect reasoning done by the rule-based inference engine. Once these cases were analyzed, we found that the incorrect reasoning occurs due to rules inconsistency that happens when several crop insect pests have symptoms in common. Regarding this issue, we are planning to add a symptom ranking mechanism i.e., when two or more crop insect pests share a symptom, this mechanism will allow to select the pest whose symptoms in common have a higher rank. The implementation of this mechanism will require the agreement of entomology experts.

We know that the current version of AgriEnt is not able to deal with crop insect pests other than those used in the case study presented above. This fact is due to the insufficiency of data and rules, which causes the inference engine to fail to provide a correct insect pest diagnosis. However, the AgriEnt-Ontology has the flexibility of extending i.e., it can be updated with more data and rules, which is difficult in a database system [11]. The next section presents our conclusions and describes future research directions which are mainly focused on addressing the issues that caused an incorrect diagnosis.

### 4.3. User-Centered Evaluation

When a new technology is introduced, it is necessary to study how users come to accept and use it. From this perspective, AgriEnt was evaluated by using the TAM (technology acceptance model) [43] which is a theoretical model that explains perceived usefulness and usage intentions. Specifically, the case study performed in this work aimed to measure the perceived usefulness, perceived ease of use, attitude, and behavioral intention to use, of the AgriEnt platform. These aspects were measured at four months postimplementation (February–May 2019). Once farmers involved in this case study used the AgriEnt Web application for insect pests' diagnosis purposes, they were asked a set of questions focused on measuring their perceptions and experiences using the platform through the aspects mentioned in the previous paragraph. The questions for perceived usefulness, perceived ease of use, and behavioral intention to use were taken from the updated TAM model known as TAM2 [43]. Meanwhile, the questions for attitude were taken and slightly adapted from [44]. The questionnaire relied on a 7-point Likert scale ranging from 1 (Strongly disagree) to 7 (Strongly agree) which allowed farmers to express how much they agree or disagree with the aspects under evaluation. A complete description of the questionnaire used in this case study is presented in Appendix A. All 20 farmers involved in this work completed the survey through the AgriEnt Web application. Then, measurement validity was evaluated in terms of reliability. Specifically, the data collected from this questionnaire were analyzed using Cronbach's alpha. The test results of this analysis are presented in Table 6.

**Table 6.** Analysis of measurement reliability: descriptive statistics and Cronbach's alphas.

| Construct | Mean | S.D. | Cronbach's Alpha |
|---|---|---|---|
| Perceived usefulness (PU) | | | 0.868 |
| PU1 | 5.60 | 0.82 | |
| PU2 | 6.05 | 0.75 | |
| PU3 | 5.85 | 0.48 | |
| PU4 | 5.95 | 0.60 | |
| Perceived ease of use (PEOU) | | | 0.806 |
| PEOU1 | 5.60 | 0.82 | |
| PEOU1 | 5.75 | 0.71 | |
| PEOU1 | 5.80 | 0.61 | |
| PEOU1 | 5.75 | 0.55 | |
| Attitude (ATT) | | | 0.864 |
| ATT1 | 5.60 | 0.82 | |
| ATT2 | 6.05 | 0.75 | |
| ATT3 | 5.80 | 0.61 | |
| Behavioral intention to use (ITU) | | | 0.829 |
| ITU1 | 5.6 | 0.82 | |
| ITU2 | 6.05 | 0.75 | |

In the first column of Table 6, the name of the constructs and the corresponding questions are listed (see Appendix A for abbreviations). The second and third columns present the mean and standard deviation respectively, while in the fourth column the values of Cronbach's alpha are shown. From the results in Table 6, we can conclude that all measurement scales have high reliability since all of them fulfill the commonly used Cronbach's alpha threshold value for acceptable reliability (0.80).

Figure 4 depicts the proposed model, which is based on TAM, as well as a summary presentation of the results. A regression analysis was performed to test this model. As can be noted from Figure 3, the results obtained support most of the causal paths postulated by the TAM theoretical model. Specifically, perceived ease of use (PEOU) significantly affects perceived usefulness (PU) with a value of 0.839. However, contrary to what the TAM model suggests, PEOU was found to have no significant effect on attitude (ATT). The effects of PU were significant in ATT. Finally, consistent with the TAM model, ATT was found to have a significant effect on the intention to use (ITU) the AgriEnt platform with a value of 0.678. This last fact suggests that attitude in farmers' acceptance of AgriEnt is relatively important and contributes to predicting the intention to use.

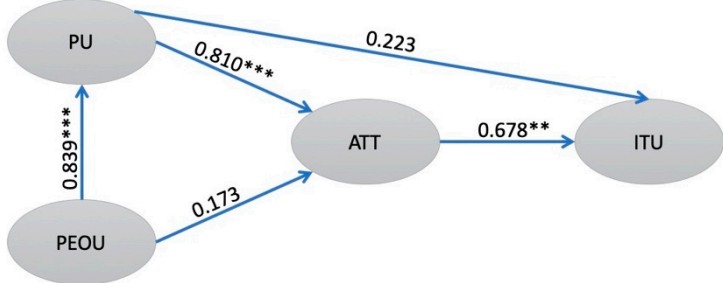

**Figure 4.** Technology acceptance model (TAM) results (** $p < 0.01$, *** $p < 0.001$).

## 5. Conclusions

Damage by insect pests is a serious challenge to agricultural producers, governments, and researchers. From this perspective, the integration of new technologies in insect pest control must be explored aiming to develop and implement safely, effectively and sustainably solutions. Based on the literature review presented in Section 2, we can conclude that there are relatively few research efforts for developing ontology-based decision support systems for crop insect pests' control and management. To address this issue, we presented AgriEnt, a Web platform that uses ontologies for decision support purposes regarding crop insect pest management. Thanks to ontologies, the expert knowledge on insect pests that affect Ecuadorian crops is available for the non-expert users to diagnoses, control, and manage insect pests. The AgriEnt platform, through the use of the ontology, has proven to produce similar answers as the professional advice given by the entomology experts involved in the evaluation process.

Although the current version of AgriEnt obtained encouraging results, it has several issues that must be addressed to provide better support to farmers. For instance, despite the AgriEnt platform is centered in a specific group of insect pests, it can be easily extended in order to deal with other insect pests from other regions. However, as more crops, diseases, and insect pests are included in the ontology, the number of rules must increase. In this sense, machine-learning methods should be investigated to discover automatically patterns in the data collected by farmers and experts and then make predictions for answering questions from the crop insect pest-management domain. On the other hand, as the current platform infers the insect pest based on the symptoms perceived by the user, this information could be inaccurate since it depends on the farmer's expertise detecting symptoms i.e., they are not able to identify specific symptoms that may affect the diagnosis accuracy. To address this issue, we plan to integrate computer vision technologies to make the symptom collections independent of the farmer's expertise i.e. using photographs of the crops to obtain more precise information about the symptoms could help to achieve a better diagnosis.

**Author Contributions:** Author Contributions: Conceptualization, K.L.-O. and R.V.-G.; methodology, M.A.P.-V. and K.L.-O.; software, K.L.-O. and J.A.G.-D.; validation, K.L.-O., J.A.G.-D., and R.V.-G.; formal analysis, M.P.S.-Z.; investigation, M.A.P.-V.; resources, R.V.-G.; writing—original draft preparation, K.L.-O.; writing—review and editing, M.A.P.-V., M.d.P.S.-Z. and R.V.-G.; visualization, K.L.-O.; supervision, R.V.-G.; project administration, R.V.-G. All authors have read and agreed to the published version of the manuscript.

**Funding:** This work has been partially supported by the Spanish National Research Agency (AEI) and the European Regional Development Fund (FEDER/ERDF) through project KBS4FIA (TIN2016-76323-R), and Seneca Foundation-the Regional Agency for Science and Technology of Murcia (Spain)- through project 20963/PI/18.

**Acknowledgments:** This work was supported by the National Technological Institute of Mexico (TecNM) and sponsored by Mexico's National Council of Science and Technology (CONACYT).

**Conflicts of Interest:** The authors declare no conflict of interest.

## Appendix A

Perceived usefulness (PU)

- PU1. Using AgriEnt improves my performance in my job.
- PU2. Using AgriEnt in my job increases my productivity.
- PU3. Using AgriEnt enhances my effectiveness in my job.
- PU4. I find AgriEnt to be useful in my job.

Perceived ease of use (PEOU)

- PEOU1. My interaction with AgriEnt is clear and understandable.
- PEOU2. Interacting with AgriEnt does not require a lot of my mental effort.
- PEOU3. I find AgriEnt to be easy to use.
- PEOU4. I find it easy to get AgriEnt to do what I want it to do it.

Attitude

- ATT1. Using AgriEnt in insect pest managing is a good idea.
- ATT2. Using AgriEnt in insect pest managing is pleasant.
- ATT3. Using AgriEnt is beneficial to insect pest management of my crops.

Behavioral intention to use (BI)

- ITU1. Assuming that I have access to AgriEnt, I intend to use it.
- ITU2. Given that I have access to AgriEnt, I predict that I would use it.

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
