# Peer review of "AgriEnt: A Knowledge-Based Web Platform for Managing Insect Pests of Field Crops"

_applsci, doi:10.3390/app10031040_

Round 1
Reviewer 1 Report
The paper introduces a web application based on knowledge and rule-based for managing insect pest of crop fields. It provides motivation for the system, discusses its architecture and implementation principles, and evaluates its performance.
Strengths
Interesting and import topic Easy to read and follow Evaluation is presentedWeaknesses
Related work should be further analyzed The scientific and technical details are too vague. Thus, the scientific challenges/ contributions are not clear.
Important issues:
There is a need to explicate the differences between the current manuscript and previous version (CITI 2018.) The related work section should analyze the challenges in applying the various systems and to indicate gaps. In its current state, it seems more like a motivation. It is beneficial to have a separate section discussing the evaluation (starting from 3.4). Also, more details are required: Who developed the rules? What were the farmers' observations (before using the system)? Since, the major content refer to the system, it might be useful to have additional insights regarding its usage, challenges, etc.
Minor:
Line 68: document-> paper
Line 151: this is not align with the paper organization (in the intro)
Line 160: the need of the figure is not clear
Line 291: the need of the figure is not clear
Reviewer 2 Report
I congratulate authors, I think this is a nice paper, easy to read. However, I have three main concerns:
1) I think authors should provide something more elaborated than figure 2 to show their ontology. Is it possible to have a graph with the whole ontology? In addition, I think authors should make their ontology public.
2) In line 358, authors state:
However, the AgriEnt-Ontology is scalable i.e., so it can be extended with more data and rules, a task that is difficult in database systems.
I disagree with this statement, so I would like authors to clarify it and provide some proof of it.
3) What was the acceptance of the application by the final users? When a new technology is introduced, it is mandatory to carry out a study of the acceptance of the new technology. The authors should use TAM, TAM2:
Venkatesh, V.; Davis, F. D. (2000), "A theoretical extension of the technology acceptance model: Four longitudinal field studies", Management Science, 46 (2), pp. 186–204.
or better UTAUT:
Venkatesh, V.; Morris; Davis; Davis (2003), "User Acceptance of Information Technology: Toward a Unified View", MIS Quarterly, 27, pp. 425–478.
Some minor errors:
In lines 18-20, authors state:
AgriEnt relies on a layered functional architecture comprising four layers: the data layer, the Semantic Web layer, the Web Services layer, and the presentation layer.
I disagree with this statement, because authors do not use the Semantic Web. According to the W3C (https://www.w3.org/2001/sw/), the Semantic Web is:
The Semantic Web provides a common framework that allows data to be shared and reused across application, enterprise, and community boundaries. It is a collaborative effort led by W3C with participation from a large number of researchers and industrial partners. It is based on the Resource Description Framework ( RDF).
The authors do not share their data on the Web or use RDF in their proposal. I think they can only state they use a "semantic layer" because they make use of an ontology.
I recommend authors to pay attention to the use of capital letters. For example, in:
the Web Services layer, and the presentation layer.
"Web Services" should be in lower case.
Or in:
such as recommender systems [8], Software Engineering [9] and Health [10–12].
"Software Engineering" and "Health" should also be in lower case.
In line 70:
implementation of SWRL-based rules for diagnosing
SWRL is later defined in line 250:
These rules were defined by using SWRL (Semantic Web Rule Language)
Acronyms should be defined first time they appear in the document.
In line 162:
application a set of symptoms he/she gathered by observing the crops he/she is in charge of
And in line 286:
the application asks users for all symptoms he/she perceived in his/her crop
Many style guides advise against the use of "he/she" and "his/her". For example, read the recommendations from APA:
https://apastyle.apa.org/style-grammar-guidelines/grammar/singular-they
It is recommended to use the singular "they" and "their".
In line 301:
This feature was necessary, so people were able to detect
I think it is better to say "This requirement was..."
In line 432-437:
12. Chen, R.-C.; Jiang, H.Q.; Huang, C.-Y.; Bau, C.-T. Clinical Decision Support System 432 for Diabetes Based on Ontology Reasoning and TOPSIS Analysis. Journal of 433 healthcare engineering 2017, 2017. 434
13. Chen, R.-C.; Jiang, H.Q.; Huang, C.-Y.; Bau, C.-T. Clinical Decision Support System 435 for Diabetes Based on Ontology Reasoning and TOPSIS Analysis. Journal of 436 Healthcare Engineering 2017, 2017, 1–14. 437
These two references are repeated.
Reviewer 3 Report
I think it is an interesting investigation and that makes a significant contribution to the area.
Round 2
Reviewer 1 Report
Thanks for addressing most of the concerns.
Two issues that can easily foxed:
improvement of writing style, especially in the new parts. IMO, the analysis of existing literature should focus on the mechanisms and not the scope. That is, what are the limitations of existing techniques,Author Response
Please see the attachment
